# Spinal Intradural Tumor Resection via Long-Segment Approaches and Clinical Long-Term Follow-Up

**DOI:** 10.3390/cancers16091782

**Published:** 2024-05-05

**Authors:** Laura Dieringer, Lea Baumgart, Laura Schwieren, Jens Gempt, Maria Wostrack, Bernhard Meyer, Vicki M. Butenschoen

**Affiliations:** 1Department of Neurosurgery, School of Medicine, Technical University of Munich, 81675 Munich, Germany; lauramichelle.dieringer@mri.tum.de (L.D.); maria.wostrack@mri.tum.de (M.W.); bernhard.meyer@tum.de (B.M.); 2Department of Neurosurgery, University Medical Center Hamburg-Eppendorf, 20251 Hamburg, Germany; l.baumgart@uke.de (L.B.); l.schwieren@uke.de (L.S.); j.gempt@uke.de (J.G.)

**Keywords:** intradural tumors, resection with extensive approaches, spinal instability, pain

## Abstract

**Simple Summary:**

Spinal intradural tumors can grow along multiple segments. Extensive approaches such as long-segment laminoplasties may be necessary to achieve the gold standard of gross total resection. Here, we present a comprehensive cohort study describing the satisfying clinical outcome of patients undergoing four or more segment approaches without requiring dorsal fixation for intradural tumor resection. The clinical outcomes are comparable to patients undergoing surgical treatment of short-segment approaches, and no patient underwent a secondary fixation for symptoms of mechanical instability.

**Abstract:**

Introduction: Spinal intradural tumors account for 15% of all CNS tumors. Typical tumor entities include ependymomas, astrocytomas, meningiomas, and neurinomas. In cases of multiple affected segments, extensive approaches may be necessary to achieve the gold standard of complete tumor resection. Methods: We performed a bicentric, retrospective cohort study of all patients equal to or older than 14 years who underwent multi-segment surgical treatment for spinal intradural tumors between 2007 and 2023 with approaches longer than four segments without instrumentation. We assessed the surgical technique and the clinical outcome regarding signs of symptomatic spinal instability. Children were excluded from our cohort. Results: In total, we analyzed 33 patients with a median age of 44 years and interquartile range IQR of 30–56 years, including the following tumors: 21 ependymomas, one subependymoma–ependymoma mixed tumor, two meningiomas, two astrocytomas, and seven patients with other entities. The median length of the approach was five spinal segments with a range of 4–14 and with the foremost localization in the cervical or thoracic spine. Laminoplasty was the most chosen approach (72.2%). The median time to follow-up was 13 months IQR (4–56 months). Comparing pre- and post-surgery outcomes, 72.2% of the patients (*n* = 24) reported pain improvement after surgery. The median modified McCormick scores pre- and post surgery were equal to II IQR (I–II) and II IQR (I–III), respectively. Discussion: We achieved satisfying results with long-segment approaches. In general, patients reported pain improvement after surgery and received similar low modified McCormick scores pre- and post surgery and did not undergo secondary dorsal fixation. Thus, we conclude that intradural tumor resection via extensive approaches does not seem to impair long-term spinal stability in our cohort.

## 1. Introduction

Intraspinal tumors make up around 15% of all CNS tumors. Among these, 40% are located intradurally and can be further subdivided into extramedullary and intramedullary tumors. Typical tumor entities comprise gliomas such as astrocytomas and ependymomas, metastases, meningiomas, hemangioblastomas, and nerve sheath tumors [1,2]. At some point, tumor growth leads to compression neural structures such as the spinal cord or nerve roots below the conus level. Back pain is often the first symptom in adult patients, whereas neurological deficits like paresis or hypesthesia mostly develop over time in dependence of the tumor localization and growth rate. The extent of these deficits can be evaluated by the modified McCormick scale (mMS) [3,4]. The primary therapy consists of surgical resection, which includes different approaches. Most commonly, interlaminar fenestration, hemilaminectomy, laminectomy, and laminoplasty are used without spinal fusion [5,6,7]. However, some tumors require extensive resection via multiple segments, i.e., equal to or above four segments. This raises the question of how tumor resection via multiple segments affects the long-term stability of the spine and the clinical outcome of our patients.

The only studies addressing this question focus on approaches for decompression in multilevel compressive cervical myelopathy. They reported that postoperative instability or kyphotic deformity is not a significant problem when using laminectomy or laminoplasty [8,9]. However, to the best of our knowledge, we did not identify any scientific data on the extensive resection of spinal intradural tumors. Therefore, we conducted a retrospective analysis to evaluate the long-term outcome with respect to clinical spinal stability and clinical outcome in patients who underwent spinal intradural tumor resection of four or more segments along the entire spinal cord. Pre-defined instability criteria included mechanical pain and neurologic deficits evaluated with the mMS preoperatively, postoperatively, and upon follow-up.

## 2. Methods

### 2.1. Study Cohort

We performed a bicentric, retrospective cohort study of all patients equal to or older than 14 years who underwent multi-segment surgical treatment for spinal intradural tumors between January 2007 and December 2023. The included centers were the University Hospital Rechts der Isar in Munich and the University Hospital Hamburg Eppendorf, both tertiary care hospitals with a high frequency of intradural pathologies. All entities of spinal intradural tumors, including tumor grading according to the World Health Organization (WHO) System, were captured. First, we selected patients who underwent spinal intradural tumor resection via an approach equal to or above four segments. In the next step, we assessed their preoperative data such as gender, age, spinal level and intra- vs. extramedullary determined by MRI (cervical vs. thoracic vs. lumbar vs. sacral spine), and type and duration of symptoms, as well as neurological and clinical status evaluated with the mMS. In the third step, we retrieved intraoperative information such as surgical approach (interlaminar fenestration, hemilaminectomy, laminectomy, laminoplasty), location of the tumor regarding the spinal cord (intradural intramedullary vs. extramedullary), and the extent of resection, i.e., a number of segments opened simultaneously. For patients with multiple surgeries due to residual tumors, tumor progress, or postsurgical complications such as hematoma, the total number of segments opened in line was assessed. Then, we analyzed the subsequent discharge reports for the presence of postsurgical pain, and both neurological and clinical status were evaluated using the Karnofsky performance status scale (KPS) and the mMS. In case of multiple surgeries, the discharge report from the last surgery was chosen. The last step comprised the analysis of the medical reports from follow-up visits for the pre-defined instability criteria. Follow-up assessment included the presence of pain and improvement compared to the preoperative condition, as well as the neurological and clinical status evaluated with mMS. The timespan to follow-up was not censored but had to be at least three months. Patients being lost to follow-up were not included in the analysis.

### 2.2. Statistics

We used Python 3.0 (Python Software foundation, 9450 SW Gemini Dr., ECM# 90772, Beaverton, OR 97008, USA) to statistically analyze the data. Mean values were reported with the 95% CI and median values with the respective interquartile range (IQR) or the minimum maximum range. We compared categorical data using the chi-square test. Multiple comparisons were made using the Kruskal–Wallis test. We analyzed the association between potential factors and permanent postoperative spinal instability (retrieved from follow-up data) using univariate regression or multivariate regression modeling with F regression. The following factors were assumed to be potentially predictive for univariate and multivariate analysis: preoperative mMS as well as the number of opened spinal segments. To assess the strength of the association, we used Spearman’s correlation coefficient. All tests performed were two-sided, and a *p*-value < 0.05 was considered statistically significant. 

### 2.3. Ethical Considerations

The presented study meets the ethical standards outlined in the Declaration of Helsinki. Also, we obtained a positive vote by a local ethics committee beforehand (number 5766/13). Due to the retrospective nature of this study, prospective patient consent was not required and was waived by our local ethics committee. 

## 3. Results

### 3.1. Patient Population

In total, we included 33 patients who underwent surgical treatment between 2007 and 2023. Among these, 57.6% (*n* = 19) were women and 42.4% (*n* = 14) were men with a median age of 44 years IQR (30–56 years) at the time of surgery. The dataset comprised 63.6% (*n* = 21) ependymomas, 6% (*n* = 2) meningiomas, and 30.4% other entities such as astrocytomas, subependymoma ependymoma mixed tumor, schwannomas, and lipomas. The tumors were mainly located in the cervical and thoracic spine (*n* = 8, 24.2%, respectively). In 18.1% of the cases, the tumor was localized in both segments (*n* = 6). In more than half of the patients (*n* = 28, 84.8%), the tumor was localized intramedullary (Table 1).

### 3.2. Surgical Approach

The most common approach of tumor resection extending over multiple segments was a laminoplasty with 66.7% (*n* = 22). Hemilaminectomy and laminectomy made up 15.1% each, while other approaches, like interlaminar fenestration, were only used in 3.1% of the cases (Table 1). The decision on the approach was taken by the senior operating surgeon based on preferences and tumor extension (laminoplasties and laminectomies were mainly chosen for intramedullary tumors, hemilaminectomy, or interlaminar fenestration applied in extramedullary tumors). Since some of the patients underwent multiple surgeries at different time points, we differentiated between the total number of segments that were opened in-line and the largest number of segments that were opened during one of the surgeries in case there were several. The median number of simultaneous opened segments during surgery, as well as the total number of opened segments, was 5, with a range of 4–14. Every patient underwent intraoperative neuromonitoring with evoked potentials in cases of spinal cord affection or electromyography in pathologies below the conus level. No patient in our study cohort received a spinal instrumentation during or after tumor resection. In summary, five patients required revision surgery for leakage (*n* = 3), postoperative hematoma evacuation (*n* = 1), or re-resection due to a tumor remnant (*n* = 1), resulting in a surgical revision rate of 15.1%. Four patients suffered from a recurrence of the tumor and needed re-resection over time with a median duration of 8 months and IQR of 3–19 months until tumor recurrence. All of the four tumor recurrences were ependymomas, WHO II.

### 3.3. Clinical Outcome

Most of the patients (78.8%) presented with mild neurological deficits before surgery with a median mMS Grade of II. mMS Grades I and II were assessed in 33.3% (*n* = 11) and 45.5% (*n* = 15), respectively (Table 2). Moderate neurological deficits (mMS Grade III) were observed in four cases, and severe deficits (mMS Grade IV) in three cases. In total, 78.8% of the patients (*n* = 26) reported back pain, neck pain, and radiculopathy. There was no significant difference in tumor expansion along the spine before surgery between patients with mild neurological deficits (mMS Grades I and II) and severe neurological deficits (mMS Grades III and IV) with a median number of five and four segments, respectively (*p* = 0.19). After surgery, 66.7% (*n* = 22) of the patients were functionally independent with mMS Grades I and II (Table 2). Nearly 18% of the study cohort (*n* = 6) with functional independence before surgery suffered from neurological deterioration immediately after surgery with postoperative mMS Grades III and IV. Two patients (6.1%) improved after surgery and regained functional independence.

All included patients had a clinical and radiological follow-up with a median time to follow-up of 13 months (IQR range, 4–55 months). Upon their last follow-up visit, 69.7% of the patients in the study cohort (*n* = 23) were functionally independent (Table 2). No improvement was observed in 12.1% of the patients (*n* = 4) exhibiting moderate to severe neurological deficits already before surgery. Moreover, one of them progressed from mMS Grades III to IV due to tumor progress. Six patients with functional independence before follow-up (18.1%) lost it permanently due to postoperative complications such as hematoma (*n* = 1), CSF leak (*n* = 1), or tumor recurrence (*n* = 2). In two cases, no plausible reason could be evaluated. One of these patients initially obtained his functional independence back after surgery but deteriorated again until his last follow-up consultation due to tumor recurrence. Another five patients (15.1%) that were functionally independent before intradural tumor resection suffered from a permanent loss of it with progression to mMS III and IV (one and four cases, respectively). In 9% of the cases (*n* = 3), functional independence was regained with mMS Grade II upon follow-up. Two of these, however, suffered from post-surgery neurological deterioration but were functionally independent with mMS Grade II before surgery (12.1%).

### 3.4. Pain Progression as a Marker of Spinal Instability

In this study, we chose mechanical back pain as one of the most important criteria for the evaluation of spinal instability. Before tumor resection, pain, including radiculopathies, nuchalgia, or other pain manifestations along the spine, was reported in 78.8% (*n* = 26) of the cases. In total, 63.6% of the cohort (*n* = 21) reported postoperative pain. However, there was no differentiation between mechanical back pain and sore pain in the discharge reports. Upon follow-up visit, 72.2% of the patients (*n* = 24) reported pain improvement. Moreover, 60.6% (*n* = 20) were entirely free of pain upon follow-up visit. In 9.1% of the cases (*n* = 3), there was neither pain prior to surgery, after surgery, or upon follow-up visit.

### 3.5. Factors Affecting Spinal Stability Concerning Change in Neurological Status

To correct for confounding factors such as the preoperative mMS, we chose the change in the mMS Grade before surgery to its follow-up value as our primary outcome measure. We sorted the outcome into three categories, namely neurological deterioration (*n* = 10 patients, 30%), stable neurological condition (*n* = 11 patients, 33.3%), and improvement in neurological condition (*n* = 12 patients, 36.3%) (Table 3). We observed no statistical difference regarding the surgical approach and lesion entity upon comparing these three categories with multivariate analysis (*p* = 0.43 and *p* = 0.25, respectively).

Furthermore, we detected no statistical difference regarding the total number of opened segments (*p* = 0.56). The median number of opened segments in the category of neurological deterioration was *n* = 4 with a range of 4–6; in the category of a stable neurological condition, *n* = 5, range (4–9); and in the category of neurological improvement, *n* = 5, range (4–14) (Table 3). 

### 3.6. Prognostic Factors

Spearman correlation analysis revealed a strong association between the preoperative mMS Grade and long-term neurological functionality (R = 0.33, *p* = 0.058). However, this result failed to reach statistical significance. The post-surgery mMS Grade was statistically significantly associated with the mMS Grade upon follow-up (R = 0.82, *p* < 0.001). Sex, age, and the total number of segments showed no association with the change in neurological status comparing neurological status before surgery and upon follow-up (*p* > 0.05).

Multivariate analysis with F regression revealed that the only statistically significant prognostic factor for the change in neurological status is the postoperative mMS Grade (*p* < 0.01) (Table 4). The number of total segments and the highest number of simultaneous open segments during surgery were not statistically significant prognostic factors for functional outcome and, thus, clinical spinal stability (*p* = 0.23 and *p* = 0.76, respectively) (Table 4).

### 3.7. Case Presentation

A young male patient initially presented in early 2019 with a distal finger flexor paresis on the left hand, hypesthesia along the C8 dermatome, and an MRC 4/5 paresis in the left leg. Together with the myelopathy, the symptoms progressed over a time span of 10 years before consultation at our outpatient care center (mMS Grade II upon first visit). A contrast-enhanced MRI scan showing an intramedullary lesion reaching from C2 to Th3 was shown (Figure 1).

The lesion was resected via laminoplasties, reaching C2 to Th3 under stable IONM (Figure 2) two weeks later. Histopathology revealed a subependymoma–ependymoma mixed tumor, WHO Grade II. After surgery, the patient suffered from a transient neurological deterioration with a finger flexor paresis 2/5 left and impairment in bladder voiding, requiring catheterization for seven days. Additionally, there was a left-sided hemi-hypesthesia, and the already known 4/5 left leg paresis remained. Nevertheless, the patient subjectively reported significant improvement one week after surgery in comparison to before. There were no complications after surgery, such as hematoma or CSF leakage. Three months after discharge, the patient received radiation therapy for 1.5 months, reaching from C1 to Th4, including the spinal canal, due to small residues of stroma on the dura. The total dosage was 50.4 Gy with five fractions of 1.8 Gy each week. After radiotherapy, the left-sided hemi-hypesthesia and 4/5 leg paresis remained. The patient maintained his neurological functionality with a Grade II mMS. After radiation, there were no signs of a tumor recurrence in the MRI scans after 6 months of follow-up. The patient was pain-free upon the follow-up visit.

## 4. Discussion

So far, there have been no studies addressing the stability of the spine after the resection of spinal intradural tumors via long-segment approaches. In our study, we showed that the resection of intradural pathologies via extensive approaches did not compromise the stability of the spine, as evaluated with the mMS Grade and mechanical pain as the primary outcome. The median number of opened segments for tumor resection was *n* = 5, with a range of 4–14. In total, 72.7% (*n* = 24) reported pain improvement, and 60.6% (*n* = 20) were free of pain upon the follow-up visit.

Functionality in everyday life is one of the key factors after spinal surgery. Impaired spinal stability causes a decrease in life quality as well as financial and social burdens for the patient. For this reason, we chose the change in the mMS as our primary outcome. Most of the included patients (*n* = 23, 69.7%) remained functionally independent with a median mMS Grade II at their last follow-up visit. Prior to surgery, functional independence was assessed in 78.8% of the patients. As many as five patients with prior functional independence lost it permanently after tumor resection. However, there was no statistically significant difference regarding the extent of tumor resection compared to the rest of the cohort, which had a median of five segments in this group as well. 

As already described in previous studies, patients with major neurological deficits prior to surgery also had a worse outcome with little or no improvement after surgery [10,11]. However, this association failed to reach statistical significance in Spearman correlation and multivariate analysis in our study. Although there is research describing tumor location, histology, and extent of resection as the most important prognostic factors for the post-surgical outcome with regard to functional independence and quality of life [12,13,14], we found that the type of surgical approach, the tumor entity, and the number of opened segments for resection are not statistically significant prognostic factors for the change in the neurological status. The only statistically significant prognostic factor in multivariate analysis for the long-term development of the neurological status and, thus, spinal stability was the postoperative mMS Grade (*p* < 0.01).

Our second criterion for the evaluation of spinal instability after intradural tumor resection over multiple segments was mechanical pain. Congruent with the literature [15], we found that tumor resection leads to pain improvement in 72.2% of our study cohort (*n* = 24). This finding supports the hypothesis that intradural tumor resection over as many as four segments and more does not impair clinical spinal stability.

Regarding the study cohort, children younger than 14 years were excluded due to their distinct higher risk of post-laminectomy kyphosis. In immature skeletal development, this is a common complication because the cartilaginous endplates are prone to compression and wedging [16]. In a study by Yasouka et al., it was demonstrated that 46% of children under the age of 15 developed spinal deformities after multi-level laminectomy. This number decreased to 6% in the age span between 15 and 24 years of age [17].

Fixation after the resection of intradural spinal tumors is a common practice especially if three or more segments are involved [18] even though there is evidence that it is only required if significant bony erosion or facet/ligamentous complex injuries occur. However, these scenarios are rare in intradural spinal tumor resection [19]. Factors potentially leading to instability after laminoplasties are preoperative deformities and performing facetectomies. None of these criteria were present in our cohort [19,20,21].

However, there are clinical and imaging follow-ups on a regular basis requiring MRI to check for tumor recurrence or progress of a possible tumor residue. In the case of fixation, there would be metal artifacts even if 1.5T MRI imaging is used. This renders it difficult to identify possible tumor recurrence or progress in an early state and imposes significant risk to the patient’s health. Therefore, laminoplasties and laminectomies without fixation after tumor resection clearly have an advantage. 

## 5. Limitations

Due to the retrospective design of this study, an analysis regarding a real causative relationship between surgery and post-surgery stability is missing. For the same reason, standardized questionnaires to evaluate neurologic deficits and pain are not available. Instead, the information was extracted from the follow-up records in outpatient care. Therefore, a uniform description of pain quality or neurological deficits before and after surgery was not available either. 

Spinal instability is associated with mechanical back pain. Sometimes, muscle spasms can be present. However, this pain is mainly elicited by bending forward or lifting heavy objects [22]. Unfortunately, in our clinical follow up, these tests are not included in the neurological examinations. Thus, pain intensity or presence can be underrated upon follow-up.

Additionally, our study’s primary outcomes are based on clinical evaluation. Clinical evaluation can be biased due to incompliance or by the patient’s or physicians’ emotional rating. Therefore, the severity of symptoms can be exaggerated or understated for primary or secondary disease gain. Objective measures such as imaging for, e.g., the evaluation of sagittal balance, were not carried out. This weakens the objectivity of the data and, therefore, generalization. Regarding the surgical approaches, we did not compare the differences between intramedullary and extramedullary resection of the lesions separately, neither laminoplasties nor laminectomies, since our primary focus lay on the number of affected segments. 

Another factor limiting the generalization of this study is the small number of patients. Even though we chose a bicentric approach, only 33 patients could be included due to the rarity of extensive spinal intradural tumors. Among these patients, we also had a limited entity variation with a bias towards ependymomas (*n* = 21). However, a low patient number and imbalance in the data set limit the statistical power and generalization of the data. Follow-up was available for all patients, but in two patients, only a short follow-up of four months was possible due to their residence abroad continuing follow-up examinations in their hometown.

## 6. Conclusions

Patients undergoing surgery for the resection of an intradural tumor over multiple segments did not seem to be at risk for impairment in long-term spinal stability in our cohort and required no secondary dorsal fixation. Even though there were cases of neurological deterioration, most of the patients reported pain improvement and kept or regained functional independence after resection, and the risk of deterioration was comparable with current evidence of short-segment laminoplasties for intradural tumor surgery. The number of segments, tumor entity, and type of approach showed no prognostic factors for spinal instability, as evaluated by our predefined criteria. The only statistically significant factor to draw conclusions for the long-term outcome was the postoperative mMS Grade.

## Figures and Tables

**Figure 1 cancers-16-01782-f001:**
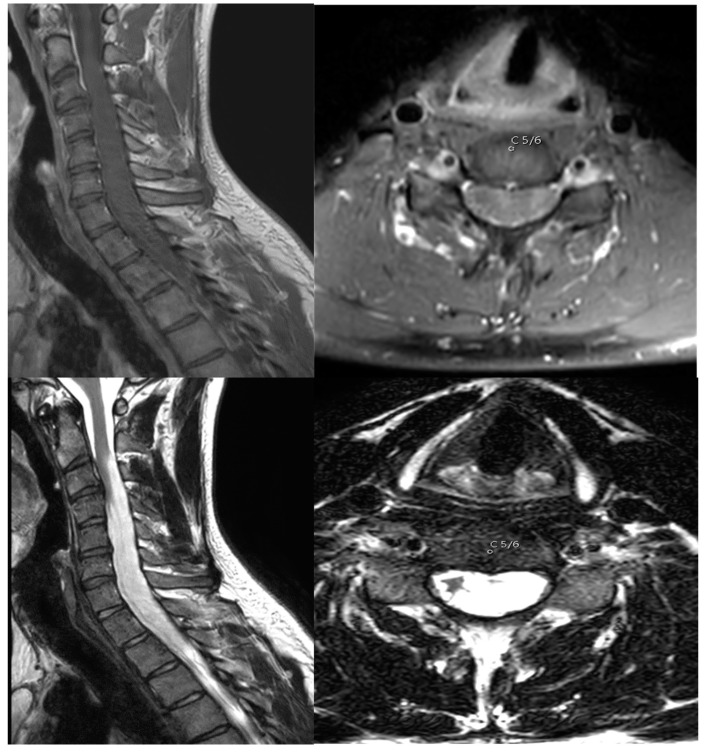
T1-MRI scan with contrast enhancement (upper image) and T2-MRI scan (lower image) showing the extent of the lesion before surgery. The axial images depict level C5/6.

**Figure 2 cancers-16-01782-f002:**
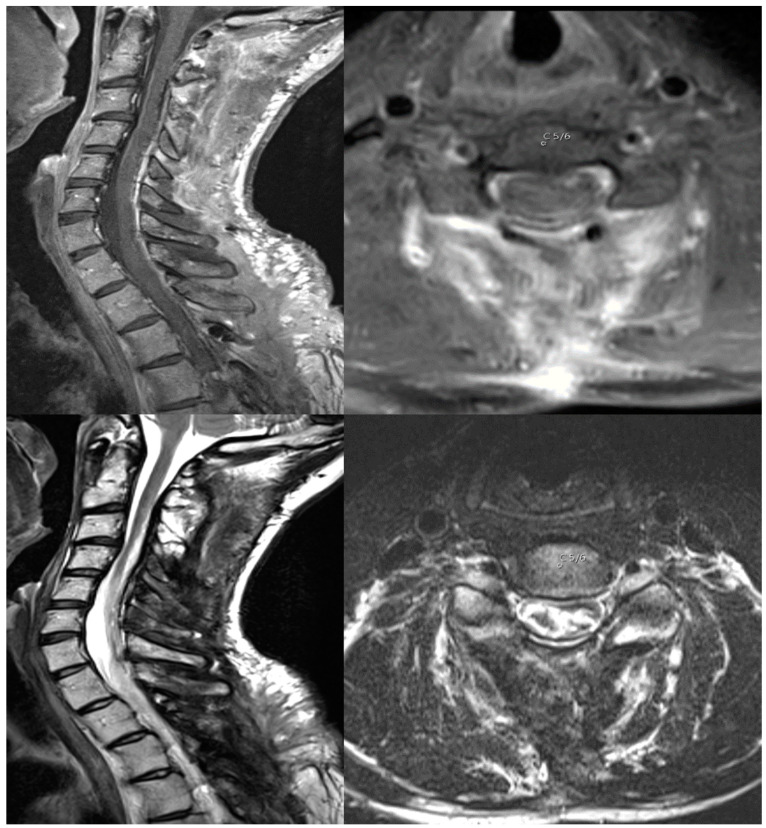
T1-MRI scan with contrast enhancement (upper image) and T2-MRI scan (lower image) showing the gross total resection of the tumor. The axial images depict level C5/6.

**Table 1 cancers-16-01782-t001:** Demographics of the cohort population.

Parameter	%/Median	Range
Female sex	57.6	
Male sex	42.4	
Age (median)	44	IQR 30–56
Number of segments	5	R 4–14
Total	5	R 4–14
Simultaneous open		
Localization		
Cervical	24.2	
CTJ	18.1	
Thoracic	24.2	
TLJ	6.1	
Lumbar	6.1	
Lumbosacral	18.1	
Ubiquitous	3.2	
Intramedullary	84.8	
Extramedullary	15.2	
Entity		
Ependymoma	63.6	
SE-E-mixed tumor	3.0	
Meningioma	6.1	
Astrocytoma	6.1	
Schwannoma	6.1	
Hemangioblastoma	3.0	
Others	15.1	
Access		
Laminoplasty	66.7	
Hemilaminectomy	15.1	
Laminectomy	15.1	
Interlaminar fenestration	3.1	
Modified McCormick Score		
Preoperative	II	
Postoperative	II	
Follow-Up	II	

*CTJ*, cervicothoracic junction; *TLJ*, thoracolumbar junction; *SE*, subependymoma; *E*, ependymoma; *IQR*, interquartile range; *R*, minimum maximum range.

**Table 2 cancers-16-01782-t002:** mMS scores prior to surgery, afterward, and upon follow-up visit.

mMS (%)	mMS Pre-Surgery	mMS Post Surgery	mMS Upon Follow-Up
I	33.3	39.4	42.4
II	45.5	27.3	27.3
III	12.1	15.2	6.1
IV	9.1	18.1	24.2

**Table 3 cancers-16-01782-t003:** Entity, access, and number of segments per category; *R* minimum maximum range.

	Deterioration	Stable	Improvement
Patients (%)	30.3	33.3	36.4
Number of segments			
Total	4 R (4–6)	5 R (4–9)	5 R (4–14)
Entity (%)			
Ependymoma	23.8	42.9	33.3
SE-E mixed tumor	0	100	0
Meningioma	100	0	0
Astrocytoma	50	0	50
Schwannoma	50	0	50
Hemangioblastoma	0	0	100
Lipoma	0	0	100
Others	33.3	33.3	33.4
Access (%)			
Laminoplasty	31.8	36.4	31.8
Hemilaminectomy	40	40	20
Laminectomy	20	20	60
Interlaminar fenestration	0	0	100

**Table 4 cancers-16-01782-t004:** Evaluation of prognostic factors with F regression for the development of the neurological and functional status.

Prognostic Factors	*p*
Sex	0.63
Age	0.25
Number of segments	
Total	0.23
Simultaneously open	0.76
Intramedullary/extramedullary	0.45
Localization	0.09
Entity	0.63
Approach	0.63
Preoperative mMS	0.37
mMS after surgery	0.002 **

** *p* < 0.005.

## Data Availability

The datasets used and/or analyzed during the current study are available from the corresponding author on reasonable request.

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
