# Peer review of "Spinal Intradural Tumor Resection via Long-Segment Approaches and Clinical Long-Term Follow-Up"

_cancers, 2024, doi:10.3390/cancers16091782_

Round 1

Reviewer 1 Report

Comments and Suggestions for Authors

The present article “Spinal Intradural Tumor Resection via Long Segment Approaches and Clinical Long-Term Follow-Up” by Dieringer et al on spinal intraductal tumor in CNS. This report includes bicentric and retrospective cohort of study on CNS tumor patients with long term follow-up. They have also mentioned the surgical procedure and clinical outcome in cohort of 33 patients. In this follow-up study, authors concluded that intradural tumor resection via extensive approaches does not seem to impair long-term spinal stability in our cohort.

Since this study focused on follow-up, I have made some suggestions, which authors should incorporate in manuscript for better understanding and representation.

·      Median Modified McCormick Scale in table form

·      Clinical outcome evaluation at long-term follow-up period, according to Odom’s criteria

·      Neurological worsening at discharge and at follow-up according to the type of the procedure (Hemilaminectomy and laminectomy) (IONM following alerts vs. non IONM)

·      Multivariate analysis at discharge and follow-up between the absence of clinical worsening and multiple variable (Odds Ratio, IC95% and p-value)

I would suggest authors to improve methodology section including MRI.

Author Response

Thank you for the opportunity to improve our manuscript according to the recommendations and the thorough work of the chosen reviewers. We incorporated all comments and think the quality improved a lot. We included information on the median modified McCormick Grades in Table 1, as suggested by Reviewer 1. Comment 3 from reviewer 1 corresponds to Table 3. We improved the methodology and included more information on the MRI use in the methods section. Unfortunately, we did not apply the Odom's criteria as it is mainly used in degenerative spinal surgery.

Reviewer 2 Report

Comments and Suggestions for Authors

The present study about the management of spinal intradural tumours using  > 4 segments approaches is well written and conducted, the methods and results sound. On the other hand few minor problems are encountered as follow:

1) the authors in the study title discuss about long-term follow-up although some patients were followed up for only 4 months (please amend, explain, discuss, clarify)

2) it would be interesting to know why almost 30% of patients did not undergo to laminoplasty that seems to be one of the most important prognostic factor to avoid instability, what were the selection criteria? (please add details, explain, argument, discuss).

3) It would be interesting to know what are the features protecting the authors patients against spinal instability (please discuss, add details, argument).

Author Response

Thank you for your valuable input. Your suggestions are included in the discussion part (for 1) and 3)) including a further limitation (follow-up) and including 3 more sources to discuss the risk of instabilities after laminoplasties (Quiceno et al, Sciubba et al. and Byvaltsev et al.).

Reviewer 3 Report

Comments and Suggestions for Authors

The authors have provoded an exhaustive and honest overview of a challenging topic – surgery of intradural tumors resected via long segment approaches together with the analysis of the long term follow up. The Abstract is well written, but in my opinion ( but it depends on the rules of the Journal) it is not generally recommended unexplained abbreviations in the Abstract text (IQR) with an exception being generally known abbreviations such as CSF, AVM,…

The Introduction part adequately addresses the importance of the problem – surgical outcomes and postsurgical instability after removal of large intradural tumors. The group of patients is quite large – this is a little surprising because of the availability of MRI. The proces of data collection and analysis is adequately described. The results are presented in the form of both text subchapters and Tables, that are quite instructive. One technical question should be asked – the removal of large tumors using interlaminar fenestration(s) is probably quite demanding due to the portions of the tumorr hidden beneath the vertebral laminae and (in the reviewer´s opinion) the problematic dural closure.  Regarding the results , not surprisingly there is a cohort of patients  that deteriorated early after surgery and also the longer follow up period (mMS IV before surgery  9.1% and in longer term follow up 24.2%) – mainly tumor recurrencies.  The problem of spinal stability after surgery is studied mainly by the aanalysis of the incidence of pain before and after surgery –seems to be rather simplifying but acceptable ( the changes suggesting spinal instability on follow up MRI after surgery maybe discussed) and the authors are aware of this limitation. The problems of possible postlaminoplasty artifacts due to the implanted fixation material is adequately discussed , but I am missing the information about the fixation technique of in laminoplasty patients used by the authors. Finally the study points to the need of earlier diagnosis and surgery of intradural tumors, because even in their experienced hands the percentage of unfavourable outcomes in the long term follw up is high.

As a conclusion I can gladly recommend the paper for publication after the answers to my querries.       

Author Response

We thank Reviewer 3 for the positive comments and added the explanation of IQR in the abstract.